# Overexpression of *PtoCYCD3;3* Promotes Growth and Causes Leaf Wrinkle and Branch Appearance in *Populus*

**DOI:** 10.3390/ijms22031288

**Published:** 2021-01-28

**Authors:** Chaonan Guan, Yuan Xue, Pengfei Jiang, Chengcheng He, Xianglin Zhuge, Ting Lan, Hailing Yang

**Affiliations:** 1College of Biological Sciences and Biotechnology, Beijing Forestry University, Beijing 100083, China; guanchaonan2014@163.com (C.G.); 18730280827@163.com (C.H.); zhugexianglin2012@163.com (X.Z.); 2State Key Laboratory of Tree Genetics and Breeding, Chinese Academy of Forestry, Beijing 100091, China; 15163816912@163.com (Y.X.); jiangpengfei0028@163.com (P.J.); 3Guangdong Provincial Key Laboratory for Plant Epigenetics, Longhua Bioindustry and Innovation Research Institute, College of Life Sciences and Oceanography, Shenzhen University, Shenzhen 518060, China

**Keywords:** CYCD3, CDK, cell division, leaf development, *Populus*, secondary growth

## Abstract

D-type cyclin (cyclin D, CYCD), combined with cyclin-dependent kinases (CDKs), participates in the regulation of cell cycle G1/S transition and plays an important role in cell division and proliferation. CYCD could affect the growth and development of herbaceous plants, such as *Arabidopsis thaliana*, by regulating the cell cycle process. However, its research in wood plants (e.g., poplar) is poor. Phylogenetic analysis showed that in *Populus trichocarpa*, *CYCD3* genes expanded to six members, namely *PtCYCD3;1–6*. *P. tomentosa CYCD3* genes were amplified based on the CDS region of *P. trichocarpa CYCD3* genes. *PtoCYCD3;3* showed the highest expression in the shoot tip, and the higher expression in young leaves among all members. Therefore, this gene was selected for further study. The overexpression of *PtoCYCD3;3* in plants demonstrated obvious morphological changes during the observation period. The leaves became enlarged and wrinkled, the stems thickened and elongated, and multiple branches were formed by the plants. Anatomical study showed that in addition to promoting the differentiation of cambium tissues and the expansion of stem vessel cells, *PtoCYCD3;3* facilitated the division of leaf adaxial epidermal cells and palisade tissue cells. Yeast two-hybrid experiment exhibited that 12 PtoCDK proteins could interact with PtoCYCD3;3, of which the strongest interaction strength was PtoCDKE;2, whereas the weakest was PtoCDKG;3. Molecular docking experiments further verified the force strength of PtoCDKE;2 and PtoCDKG;3 with PtoCYCD3;3. In summary, these results indicated that the overexpression of *PtoCYCD3;3* significantly promoted the vegetative growth of *Populus*, and PtoCYCD3;3 may interact with different types of CDK proteins to regulate cell cycle processes.

## 1. Introduction

Cyclin is a key regulator of cell cycle progression. Cyclins have been classified into several classes on the basis of sequence similarity, expression pattern, and protein activity during the cell cycle [1]. Among them, plant D-type cyclin (CYCD) cDNAs were first isolated from *Arabidopsis* [2] and alfalfa [3] because of their ability to functionally complement yeast strains defective in G1 cyclin-deficiency. These cDNAs are defined as CYCDs in accordance with the sequence homology with animal CYCD and the presence of conserved LxCxE (where x represents any amino acid) motif [4,5,6]. Each class of cyclin is divided into several subclasses, and many family members exist in each subclass. For example, in *Arabidopsis* CYCD, the CYCDs are divided into seven subclasses (*CYCD1–7*), of which *CYCD3* has three members, namely, *CYCD3;1–3* [1]. The kinase activity present in heterodimers formed by cyclins and cyclin-dependent kinases (CDKs) is a major cell cycle regulator [7]. The binding site of cyclin and CDK is approximately 100 amino acids long and called the cyclin box [8]. The different domains that bind to cyclin are used as the standard for CDK classification. Different types of CDK contain different conserved domains [9,10]. *CYCD* is considered to regulate the G1-to-S transition [11]. Many growth factors, such as cytokinin, auxin, brassinolide, and sucrose, regulate the expression of *CYCD* [12]. When stimulated by environmental factors, CYCD combines with CDK to form an inactive CYCD/CDK complex. This complex is activated by CDK-activating kinase (CAK). Then, it stimulates the expression of S-phase related genes via the RBR/E2F/DP pathway [12].

The overexpression of *AtCYCD3;1* increased the number of epidermal cells on the adaxial epidermis of leaves but failed to form apparent spongy tissue and palisade tissue [13]. The overexpression of *AtCYCD3* in tobacco altered the structure of the apical meristem and accelerated the initiation of leaf development [14]. The transgenic plants that overexpressed *NtCYCD3;4* not only grew faster than wild-type plants but also had smaller cells in young leaves and reduced cell numbers [15]. These findings demonstrated that many plant *CYCD3* members are involved in the differentiation of meristems, and they influence the development of leaves. Analysis of *Arabidopsis CYCD3* triple loss of function mutants showed that the stem and hypocotyl diameters of the mutants were significantly reduced due to the decreased mitotic activity of the cambium cells [16]. Mutants *cycd3;1* and *cycd3;1-3* showed thinner roots than the wild type, and the number of vascular cells in plants of mutant *cycd3;1* was substantially decreased [17]. These functional loss experiments confirmed that *Arabidopsis CYCD3* affects the differentiation and proliferation of vascular cambium cells. Although *CYCD* plays a vital role in the cell cycle process, the research involving CYCD function has only been conducted in herbaceous plants, such as *Arabidopsis* and tobacco, and few studies have reported on the function of *CYCD* in woody plants [18]. In woody plant stems, the activation of cambium cells is related to the thickness of the stem [19], and leaf development is directly related to plant photosynthesis and vegetative growth [20]. Therefore, it has a great theoretical and application value to study the function of *CYCD3* in woody plants.

*Populus tomentosa* is a widely cultivated tree species of economic importance in China. A relatively complete system of genetic transformation has been developed in *P. tomentosa*, which makes this species one of the most significant tree species for forest genetic research [21]. Six homologous *CYCD3* genes were examined in this study to document their expression and role in *Populus trichocarpa*. Five *PtoCYCD3* genes were identified and amplified in *P. tomentosa* based on the CDS region of *P. trichocarpa CYCD3* genes. *PtoCYCD3;3* was selected as the target gene by examining the gene expression patterns in various tissues. Phenotypic analysis of the overexpression of *PtoCYCD3;3* in *P. tomentosa* found that *PtoCYCD3;3* not only promoted leaf cell division but also increased the thickness of secondary xylem and secondary phloem by increasing cambium cell activity, thereby promoting vegetative growth of plants. In this study, the effects of *PtoCYCD3;3* on *Populus* growth and development were examined and CDK family members that interact with PtoCYCD3;3 were analyzed, laying a foundation for further exploration of the molecular mechanism of cyclins in woody plants.

## 2. Results

### 2.1. Identification of Members of CYCD3 Subtype and Analysis of Tissue Expression Patterns in P. tomentosa

A total of 24 full-length *CYCD* candidate genes were obtained via searching homologies in the *P. trichocarpa* genome by TblastN, using *Arabidopsis CYCD* genes as a template. CDD domain detection showed that most of the genes have typical and complete CYCD conserved domains, while *PtCYCD1;1*, *PtCYCD5;3* and *PtCYCD6;2* were partially misannotated with truncated protein structures. Thus, we manually revised these sequences based on poplar EST database in NCBI, or the sequences of their phylogenetic closet paralogs (Appendix A). Phylogenetic analysis between them and the *Arabidopsis CYCD* family showed that these poplar *CYCD* genes and *Arabidopsis* clustered into six large branches (Figure 1A marked with red dots), and each branch had a high support rate. The branching of the gene tree closely resembled the pattern of the subtype classification of *Arabidopsis CYCD*, suggesting that the *Populus CYCD* was also mainly divided into six different subtypes, namely *CYCD1*, *CYCD2/4*, *CYCD3*, *CYCD5*, *CYCD6*, and *CYCD7*. These candidate genes were named on the basis of the classification and the location of the genes in the genome of *P. trichocarpa*. The phylogenetic tree revealed that some CYCD subtypes expanded independently after the divergence of *Populus* and *Arabidopsis*. For example, *CYCD1*, *CYCD3*, and *CYCD6* were the largest subtypes in *P. trichocarpa*, containing seven, six, and five members, respectively, whereas those in *Arabidopsis* contains one, three, and one member, respectively. The number of amino acid residues encoded by the *CYCD* of *P. trichocarpa* was between 257 and 386.

Six *CYCD3* genes were examined for homologous patterns of nucleotides in *P. trichocarpa*. Five *PtoCYCD3* genes were identified and amplified in *P. tomentosa* based on the CDS region of *P. trichocarpa CYCD3* genes. Among them, *PtoCYCD3;4* was not amplified. The expression levels of six different tissues of *P. tomentosa* under normal growth conditions were detected using RT-qPCR to analyze the tissue expression pattern of *PtoCYCD3*. The results showed that the five *PtoCYCD3* members exhibited divergent temporal and spatial expression patterns (Figure 1B). *PtoCYCD3;1*, *PtoCYCD3;5* and *PtoCYCD3;6* had the highest expression in the young leaf, *PtoCYCD3;2* in the mature leaf, while *PtoCYCD3;3* in shoot tip, respectively. We also integrated the transcript profiles of *CYCD3* type homologous genes previously reported in *Populus tremula* and *Populus trichocarpa* [22,23]. In *Populus trichocarpa*, all the members also showed differential expression patterns, but the similarity is that all showed high or relatively high expression in young leaf (Appendix A). In *Populus tremula*, homologous genes of *PtoCYCD3;3*, *PtoCYCD3;5* and *PtoCYCD3;6* were detected in the apical meristem regions of stem, shoot and root (Appendix A). All together, *PtoCYCD3;3* and its homologs were generally detected in the meristems in the poplar species. Thus, it was selected as our research target to explore its regulation for poplar growth and development.

### 2.2. Overexpression of PtoCYCD3;3 Promotes the Vegetative Growth of Populus

*35S::PtoCYCD3;3* was transformed into *P. tomentosa* via the Agrobacterium-mediated leaf disc method, and ten transgenic lines were used to measure growth parameters (Figure 2). We randomly selected three lines of the 10 transgenic lines for expression measurement. Three lines of *Pro35S::PtoCYCD3;3* were measured with increased expression levels (Figure 2E). The growth parameters of these three transgenic lines were listed in Appendix A. The phenotype of the overexpressed plants intensively changed during the morphological observation period compared to the wild-type plants, with increased plant height, branching, thickened stems, and enlarged leaves and wrinkles (Figure 2A). The phenotypic effect of *PtoCYCD3;3* on *Populus* was characterized by measuring the plant height, stem thickness, internode number, internode length, and leaf length and width at 70, 100, 130, and 170 days after potting. That the plant height of the overexpressed plants was significantly higher than that of the wild-type plants during the growth process from 70 days to 170 days (*p* < 0.01) (Figure 2H) may be due to the increase in leaf cell number or the increase in the cell area. The measurement results of the number and length of the internode showed that the number of internodes of the overexpressed plants was significantly higher than that of the wild-type plants (*p* < 0.01) (Figure 2I) and the internode length of the overexpressed plants was significantly longer than that of the wild-type plants (*p* < 0.05) (Figure 2J). This finding showed that the increase in plant height of the overexpressed plants was caused by the increase in the number and length of the internode. The overexpressed plants began to branch at 100 days after potting. Each plant had 3–5 branches at 170 days, whereas branches never appeared in the wild-type plants (Figure 2A and Appendix A). The leaf length, leaf width, and leaf area of mature leaves were measured to detect whether the leaf size changed (Figure 2B,D,G). The results showed that the leaf length and leaf width of the overexpressed plants increased significantly (*p* < 0.05) (Figure 2G). The leaf area increased to 163% ± 7%. The stem thickness of the overexpressed plants increased considerably (*p* < 0.05) (Figure 2C,F). These results all show that overexpression of *PtoCYCD3;3* could significantly promote the vegetative growth of *Populus*.

### 2.3. Effect of Overexpression of PtoCYCD3;3 on the Growth and Development of Poplar Leaves

The leaf area of the overexpressed plants was larger than that of the wild-type plants, may be due to the increase in leaf cell number or the increase in the cell area. The number of cells per unit area in the three sections of the adaxial epidermis of the young leaf and mature leaf was counted, including leaf base, leaf middle, and leaf tip. The results showed that in young leaves, no significant difference was found in cell number and cell size between the overexpressed plants and the wild-type plants (Figure 3A,B and Appendix A). However, in mature leaves, the cell number in the overexpressed plants was significantly more than that of the wild-type plants (*p* < 0.01) (Figure 3C,D), and the cell size was significantly smaller (*p* < 0.05) (Appendix A). This finding indicated that the increase in cell number, not the increase in the cell area, was the reason why the transgenic plants had larger leaves than wild-type plants during the development of young leaves to mature leaves. Moreover, the cell shape factor [13] per unit area of the leaf base, middle, and tip on the adaxial epidermis cells of young and mature leaves were detected. The results showed that the morphology of young and mature leaf cells of the overexpressed plants did not differ from that of the wild-type plants (Appendix A). These results suggested that the overexpression of *PtoCYCD3;3* promoted the division of adaxial epidermis cells of poplar leaves but did not affect the cell morphology.

Plant leaves are usually composed of leaf epidermis, palisade tissue, and spongy tissue. In this study, the longitudinal section of the leaves was observed to detect cell changes in different tissues in the leaves. The number of palisade layers of the overexpressed plants did not change compared with that of the wild-type plants, but its thickness was significantly narrower (*p* < 0.01) (Figure 3E,F), whereas the spongy tissue thickness did not change (Figure 3F). The thickness of the leaf cut surface was significantly narrowed (*p* < 0.01) (Figure 3F). This result showed that the smaller cell of the palisade tissue led to the narrowing of the leaf cut surface. Meanwhile, the number and area of palisade tissue cells were detected. The results showed that the number of palisade cells per unit area of the leaves of the overexpressed plants was significantly increased compared with that of the wild-type plants (*p* < 0.01), and the cell area was significantly smaller (*p* < 0.01) (Figure 3H,G). These results indicated that the overexpression of *PtoCYCD3;3* promoted the division of the palisade tissue cells in poplar leaf.

In green plants, palisade tissue is the main place for leaf photosynthesis [20]. Photosynthetic electron transfer is completed by the cooperation of two photoreaction systems. Therefore, the electron transfer rate (ETR) of the leaves of the overexpressed and wild-type plants under different light intensities and the net assimilation rate (NAR) under specific light intensities were tested. The results showed that the overall change trend of the electron transfer rate of photosystem I and photosystem II in the overexpressed and wild-type plants under different light intensities was similar. The electron transfer rate raised linearly with the increase in light intensity. The increase in electron transfer rate gradually slowed down when it was close to the light saturation point. The electron transfer rate was basically stable when the light saturation point was reached. However, at the light saturation point, the maximum electron transfer rate of the overexpressed plants was considerably lower than that of the wild-type plants (*p* < 0.05) (Figure 4A,B). The light intensity (737 μmol × m^−2^ × s^−1^) was selected after poplars reached the light saturation point to determine the net assimilation rate per unit area of the overexpressed plants. The results showed that the net assimilation rate per unit area of these plants was significantly lower than that of the wild-type plants (*p* < 0.01) (Figure 4C). However, the net assimilation rate of the whole leaf was higher than that of the wild-type plants due to the increase in the leaf area of the overexpressed plants (*p* < 0.01) (Figure 4D).

### 2.4. Overexpression of PtoCYCD3;3 Promotes Secondary Growth of Populus Stems

Morphological experiments showed that the stem thickness of the overexpressed *PtoCYCD3;3* plants increased dramatically. The vascular tissues of the second, 10th, and 20th internode of the overexpressed and wild-type plants were observed via microdissection to further clarify the influence of *PtoCYCD3;3* on the radial growth of *Populus* stems. The young stems (second internode) of the overexpression plants did not undergo morphological changes compared with those of the wild-type plants (Figure 5A and Appendix A). The widths of the secondary phloem and secondary xylem of the 10th and 20th internode of the overexpressed plants significantly increased relative to those of the wild-type plants (*p* < 0.01) (Figure 5B,D,F,G), and the area of vessel cells was considerably bigger (*p* < 0.05) (Figure 5H).

Vascular cambium differentiates inwardly into secondary xylem and outwardly into secondary phloem through periclinal division. The cambium and its directly produced daughter cells are called cambium-derived tissues, which could be used to measure the activity of the cambium [24]. Statistics on the widths of the stem cambium-derived tissues of the 10th and 20th internode showed that the width of the cambium-derived tissues of the overexpressed plants was substantially increased compared with that of the wild-type plants (*p* < 0.05) (Figure 5C,E–G). This result indicated that the overexpression of *PtoCYCD3;3* increased the activity of cambium cells in *Populus* stems and facilitated the differentiation of cambium cells into secondary xylem and secondary phloem, thus leading to the proliferation of secondary vascular tissue cells. Furthermore, the overexpression of *PtoCYCD3;3* significantly resulted in the expansion of vessel cells. In short, the secondary growth of *Populus* stems was increased by the overexpression of *PtoCYCD3;3*, ultimately contributing to an increase in stem growth.

### 2.5. Identification and Structural Analysis of Members of the CDK Family of P. tomentosa

The overexpression of *PtoCYCD3;3* in *P. tomentosa* produced an obvious phenotypic difference, and the D-type cyclin needed to form an active complex with CDK to play its role, so the interaction between PtoCYCD3;3 and CDKs was not known. Eighteen *CDK* genes were searched in the *P. trichocarpa* genome. Seventeen *CDK* genes were amplified in *P. tomentosa* based on the CDS region of *P. trichocarpa CDK* genes. Among them, *PtoCDKG;2* was not amplified. These genes all had typical CDK conserved domains and a phylogenetic relationship with *Arabidopsis CDK* genes (Figure 6A). Phylogenetic analysis revealed seven different subtypes of these *CDK* genes, which were classified into *CDKA*, *CDKB*, *CDKC*, *CDKD*, *CDKE*, *CDKF*, and *CDKG*. The *Arabidopsis CDK* family has 14 members, of which *CDKE* and *CDKG* expanded in *Populus*. In *Arabidopsis*, these two subtypes contained one and two members, respectively; in poplar, they included two and four members, respectively.

The catalytic center of CDK includes a 300 amino-acid core with a typical folding form of protein kinase [25]. The domain includes a two-lobe structure connected by a hinge region, an amino-terminal lobe, and a carboxy-terminal lobe. The amino-terminal lobe contains five β-sheets and one α-helix (C-helix) [25]. The C-helix binds to the synergistic cyclin and helps determine the specificity of CYC-CDK binding (Figure 8). The conserved domain that binds to cyclin located on the C-helix was used as the standard for CDK classification (Figure 6B). Analysis of the sequence of 17 *CDK* family members of poplar with the *Arabidopsis* homologous proteins revealed that while the *CDK* subtype members in poplar expanded, their conserved domains were highly consistent with their orthologous proteins in *Arabidopsis* (Figure 6A,B). Among them, CDKA had PSTAIRE, CDKB had PPTALRE or PPTTLRE, CDKC had PITAIRE, CDKE had SPTAIRE, and CDKG had PLTSLRE [10].

### 2.6. In-Vitro Interaction Analysis between PtoCYCD3;3 and Members of the Poplar CDK Family

Yeast two-hybrid experiment was used to detect the interaction between PtoCYCD3;3 and 17 PtoCDK proteins and explore which CDK proteins interacted with PtoCYCD3;3. First, PtoCYCD3;3 and PtoCDK proteins connected to pGBKT7 and pGADT7 vectors, respectively, to form the fusion protein, and nine PtoCDK proteins (i.e., PtoCDKB1;2, PtoCDKB2;1, PtoCDKC;2, PtoCDKD;1, PtoCDKD;2, PtoCDKE;2, PtoCDKF;1, PtoCDKG;3, and PtoCDKG;4) interacted with PtoCYCD3;3 through screening of defective medium (Figure 7A). For non-interacting protein pairs of interchange expression vectors, the screening of defective medium found that PtoCDKA;1, PtoCDKB1;1, and PtoCDKB2;2 also interacted with PtoCYCD3;3 (Figure 7B). These results showed that 12 CDK proteins interacted with PtoCYCD3;3, all of which were able to grow on the corresponding defective medium, but their growth rate and colony color were distinct. The intensity of the interaction was speculated using its growth rate and color depth on the defective medium as observational indicators. Among them, PtoCDKE;2 had the strongest interaction relationship with PtoCYCD3;3, whereas PtoCDKG;3 had the weakest (Figure 7C).

Molecular docking experiment analysis of the PtoCYCD3;3-PtoCDKE;2 and PtoCYCD3;3-PtoCDKG;3 complexes showed that the docking scores of the two protein complexes were −74.51 and −70.18, respectively (Appendix A). The more negative the docking score is, the better the protein–protein binding. The interaction of PtoCDKE;2 with PtoCYCD3;3 was stronger than that of PtoCDKG;3 with PtoCYCD3;3. This finding proved the reliability of the result of the Y2H experiment. The docking studies indicated that R7, S8, S9, E77, and L86, the residues in PtoCDKE;2, were involved in binding with R107, E158, T160, and R303 in PtoCYCD3;3 through salt bridge and hydrogen bond interactions (Figure 8A,B). The residues in PtoCDKG;3, E430, R431, K449, D501, and K583 were also involved in binding with E69, K76, T160, E178, and N198 in PtoCYCD3;3 through salt bridge and hydrogen bond interactions (Figure 8C,D). The binding mode of the CYCD3/CDK complex was demonstrated by molecular docking, in which the Thr160 site of CDK was not phosphorylated and had no kinase activity. Meanwhile, the T160 of PtoCYCD3;3 was easily combined with PtoCDK in the form of hydrogen bonds, thus playing an important role (Figure 8).

## 3. Discussion

The *PtoCYCD3;3* overexpressed poplars became taller than the wild-type plants, the number of internodes increased, the length of internodes elongated, the stem thickness increased, and the leaf size increased. The number of cells per unit area increased and the cell area decreased in the adaxial epidermis and palisade tissue cells, indicating that the overexpression of *PtoCYCD3;3* affected the leaf development by promoting leaf cell division. Studies have shown that overexpressing *NtCYCD3;4* in tobacco displayed decreased cell size, no change in morphology, and increased growth rate in young leaves relative to wild-type plants [15]. The *Arabidopsis cycd3;1-3* mutant showed decreased cell number and increased cell size in adaxial epidermis cells and palisade tissue cells [26]. These findings indicated that *CYCD3* functions in a manner similar to that of herbaceous plants for the cell division of leaves in woody plants. The decrease in net photosynthesis rate per unit area was presumed to be caused by a decrease in the number of chloroplasts caused by the reduction in palisade tissue cells. The number of chloroplasts is directly controlled by cell size, and the promotion of chloroplast proliferation depends on the enhancement of cell expansion after mitosis [27,28]. In this study, although the net photosynthesis rate per unit area of overexpressed plants decreased, their total photosynthesis intensity increased due to the increase in the total area of their leaves. A significant increase in stem thickness and the expansion of vessel cells in overexpressed *PtoCYCD3;3* poplars was apparently related to vigorous cambium cell activity that may have enhanced the differentiation and development of secondary xylem and secondary phloem. Schrader et al. found that *PtCYCD3;3* was highly expressed in the cambium region [22]. Taken together, the results showed that *PtoCYCD3;3* could promote the differentiation of stem cambium cells and leaf development and increase biomass by enhancing plant vegetative growth.

Although the understanding of plant cell cycle regulation increased, little is known about the composition and spatiotemporal occurrence of various CDK/CYC complexes [29]. In *Arabidopsis*, four CDKs interact with AtCYCD3, including AtCDKA;1, AtCDKB1;1, AtCDKB2;1, and AtCDKE;1 [29]. In the present study, 12 PtoCDKs could interact with PtoCYCD3;3 through the preliminary screening of Y2H technology. These PtoCDKs were distributed into seven CDK subtypes. This finding may imply that *PtoCYCD3;3* functioned more extensively in *P. tomentosa*. Amino acid sequence analysis showed that the conserved amino acid residues in the C-helix of PtoCDKE;2 and PtoCDKG;3 that interacted with cyclin were located at positions 65–71 and 470–476 of the primary amino acid sequence, respectively. Molecular docking suggested that the amino acids near the conserved domains of PtoCDKE;2 and PtoCDKG;3 could interact with the amino acids in the cyclin box (Appendix A) of PtoCYCD3;3, mainly through hydrogen bonds and salt bridges. Cyclin interacts with CDK and forms transient or stable complexes that are essential for cell cycle progression [30]. Therefore, to understand the dynamics and regulation of cell cycle, the systematic identification and characterization of these protein complexes are important.

As an important factor mediating the regulation of endogenous and exogenous developmental signals on cell division, CYCD is co-regulated by a variety of plant hormones such as Auxin, cytokinin and brassinolide [31,32,33]. Constitutive expression of *CYCD3* in transgenic plants allowed induction and maintenance of cell division in the absence of exogenous cytokinin [23]. When the wild-type and *35S::PtoCYCD3;3* plants were 4 months old, quantitative analysis of hormones showed that the content of some hormones that promote cell growth and division decreased, including indole-3-acetic acid (IAA) and trans-zeatin (TZR) (Appendix A), whereas the content of abscisic acid (ABA) that inhibited plant growth increased (Appendix A). The idea of a negative feedback loop controlling cell division at the cambium and in branching is mentioned in common between Arabidopsis and Populus secondary growth involving auxin and cytokinin [34,35].In our paper, reduction of endogenous cell division and auxin content in stem and leaf resulting from the overexpression of *PtoCYCD3;3* might be another good example of a negative feedback loop. The brassinolide (BR) content in the leaves and stems of the overexpressed plants was significantly higher than that of the wild-type plants (*p* < 0.05) (Appendix A). The increase in BR content may be a positive feedback regulation mechanism of plants or because BR and the *PtoCYCD3;3* signaling pathways are consistent. Studies have shown that the expression of *CYCD3* could be positively regulated by BR [31]. Furthermore, BRs regulate the differentiation of vascular tissues in the shoot, considering *Arabidopsis* and rice (*Oryza sativa*) BR loss-of-function mutants have less xylem [36,37,38]. In BR-deficient mutants, the mitotic activity of roots and leaves was restored by the overexpression of the cell cycle gene *CYCD3;1* [39,40]. The content of gibberellin (GA3) in stems increased (Appendix A) but decreased in leaves, suggesting that plant stems and leaves have different hormonal regulation modes. A complex network involving hormones and transcription factors controls the development of axillary buds into lateral branches. Auxin, especially IAA is the main representative of this class of plant hormones, and it is involved in the regulation of branches [41]. The growth of lateral buds could be hindered by auxin [42]. In this study, the IAA content of the leaves and stems of the overexpressed plants was significantly lower than that of the wild-type plants. The decrease in auxin concentration may relieve the growth inhibition of axillary buds for the overexpressed plants to branch during the morphological observation period. Strigolactone (SL) inhibit shoot branching through the *max* dependent pathway in Arabidopsis [43] and the cambium activity of all *max* mutants in Arabidopsis is decreased [44]. In this study, overexpression of *PtoCYCD3;3* increased the cambium activity in the stem and produced branching (Figure 2 and Figure 5), which indicates that there may be some relation between *PtoCYCD3;3* and SL. The key factor that made up the canopy was branching, which reflected the main characteristics of the canopy structure. Highly branched plants are considered to have an improved leaf area index, which increases light capture efficiency and photosynthesis, thus leading to increased biomass production [45,46,47]. A relatively complicated interaction network exists between hormones, and further study is needed for the nodes of hormone interaction.

In the present study, the overexpression of *PtoCYCD3;3* promoted the cell cycle G1/S transition of perennial poplars, thus accelerating the cell division of poplar meristem, such as apical buds, axillary buds, and stem cambium. It also accelerated plant vegetative growth and produced obvious branches. This finding provided research ideas and economic value for forest genetic improvement through increasing the wood yield of poplars by increasing plant biomass.

## 4. Materials and Methods

### 4.1. Gene Identification and Cloning 

TBLASTN searches were performed with default algorithm parameters in Phytozome against the *P. trichocarpa* v3.0 genomes (https://phytozome.jgi.doe.gov/pz/portal.html), by using 51 cyclin protein sequences of Arabidopsis as queries to identify the *CYCD* genes in *Populus*. We also referenced a study in Menges et al., 2007, to ensure no genes were missed in our study (Appendix A and Appendix A). All potential candidates identified were examined using the NCBI Conserved Domain Database (CDD) (https://www.ncbi.nlm.nih.gov/Structure/cdd/wrpsb.cgi?) to confirm the presence of typical Cyclin N- and C-terminal domains in the protein structures. A few of the members were partially misannotated during the automated genome annotation process. Thus, we performed manual reannotation using EST database or their phylogenetic closet paralogs (Appendix A). Full-length protein sequences were aligned using the MUSCLE online service. The phylogenetic tree was established utilizing the maximum-likelihood (ML) method in PHYML software [48] with Whelan and Goldman (WAG) amino acid substitution model. The proportion of invariable sites (I) and gamma distribution (G) parameter was estimated, and the bootstrap replicates were set to 1000. AtCYCA and AtCYCB proteins were used as an outgroup. TBLASTN searches were performed with default algorithm parameters in Phytozome against the *P. trichocarpa* v3.0 genomes, by using 14 CDK protein sequences of *Arabidopsis* as queries to identify the *CDK* genes in *Populus*. The identification method of *CDK* genes in *P. tomentosa* was similar to that of *CYCD* genes.

*PtoCYCD3* and *PtoCDK* genes were cloned from the cDNA of *P. tomentosa* leaf by using the primers listed in Appendix A. The primers were designed in accordance with the reference genomic sequence of *P. trichocarpa.* The PCR products were cloned into the pEASY-Blunt vector (TransGen, Beijing, China) and sequenced. The CDS region of genes were isolated. The sequence of *PtoCYCD3* and *PtoCDK* genes were listed in Appendix A.

### 4.2. Construct Generation and Genetic Transformation

The CDS of *PtoCYCD3;3* was inserted into the *pBI121OB* vector (Appendix A) between Bam HI and Xho I enzyme sites to generate the overexpression construct *35S::PtoCYCD3;3*. Primer sequences were listed in Appendix A. The construct was transformed into *P. tomentosa* via Agrobacterium-mediated transformation. The leaves were infected with *Agrobacterium tumefaciens strains* LBA4404 carrying the recombinant plasmids. Subsequently, the leaves were cultivated on MS medium (pH 6.0) containing 100 μmol/L acetosyringone, 0.1mg/L 1-naphthaleneacetic acid (NAA), 1.0 mg/L N-(Phenylmethyl)-9H-purin-6-amine (6-BA), 3% (w/v) sucrose, and 0.6% (w/v) agar for 48 h in the dark. These explants were then transferred to selective medium (MS supplemented with 0.1 mg/L NAA, 1.0mg/L 6-BA, 25mg/L kanamycin, 200mg/L cefotaxime, 3% (w/v) sucrose, and 0.6%(w/v) agar). These explants cultivated approximately eight weeks on selective medium under light, during which the medium was refreshed every two weeks. The generated shoots were cut from calli and transferred to rooting medium (WPM supplemented with 0.1 mg/L 3-indolebutyric acid, 25mg/L kanamycin, 200 mg/L cefotaxime, 1.5% (w/v) sucrose, and 0.6% (w/v) agar). At least 10 individual transgenic lines were generated for the construct.

### 4.3. Plant Materials and Growth Conditions

*Populus tomentosa Carr.* (Clone 741) was used as the wild type. During the genetic transformation experiment, *P. tomentosa* was grown in the greenhouse at 25 °C under a 14:10–h light: dark cycle. The transgenic and wild-type plants were grown in an experimental field (Beijing, China: 39°99′N, 116°22′E) during the growing season for 6 months. Morphological features, including plant height, internode numbers, internode length, and stem diameter, were continuously measured for 170 days with 10 transgenic lines. The size of mature leaves was measured at 130 days. ImageJ was used to calculate the leaf area. All statistical analyses were performed using SPSS v16.0. Independent-Samples T-test was used.

### 4.4. RNAs Extraction, RT-qPCR and Expression Analysis

Shoot apical buds (~5 mm in length) were collected as shoot tip tissues. Young leaf, mature leaf, secondary xylem, and secondary phloem (including cambium, 20th internode counting from the tip) and root tissues were harvested from three independent 5-month-old wild-type *P. tomentosa* grown in a phytotron (under light and dark cycle of 16 and 8 h, respectively). After peeling off the bark of the stem segment (20th internode), phloem and xylem tissues were collected by scraping the inner surface of the bark and outer surface of the wood with a razor blade. All samples were immediately frozen in liquid nitrogen and stored at −80 °C until use.

RNAs were extracted from the samples above via RNAprep Pure Plant Kit (Polysaccharides and Polyphenolics-rich) (DP441, TIANGEN, Beijing, China) and subjected to cDNA synthesis by using PrimeScript^TM^ RT Reagent Kit (RR037A, TaKaRa, Japan). *PtoCYCD3* transcript levels were determined via real−time quantitative polymerase chain reaction (RT-qPCR) by using TB Green^TM^ Premix Ex Taq^TM^ II (Tli RNaseH Plus) (RR820A, TaKaRa, Japan) and analyzed via the 2^−ΔΔCt^ method with the *Populus*
*Actin* (GenBank number XM_002316253) housekeeping gene. Three technical repeats were performed for each pair of primers. All primers used for RT-qPCR are listed in Appendix A.

The expression profiles of *CYCD3* type genes in *Populus tremula* and *Populus trichocarpa* were bulit using the published data in Schrader et al. and poplar eFP Browser, respectively [22,23].

### 4.5. Scanning Electron Microscopy (SEM)

The leaf tips, leaf middle, leaf base of the young leaves (third leaf counting from the tip), and the mature leaves (the 9th leaf) were taken from 4-month-old wild-type and *35S::PtoCYCD3;3* plants. The samples (0.2–0.3 cm) were fixed in FAA solution (70% ethanol: glacial acetic acid: formaldehyde = 18:1:1) for 24 h at 4 °C. Subsequently, the samples were dehydrated in graded ethanol series (70%, 80%, 90%, 95%, and 100% ethanol) for 15 min, and then incubated at 4 °C overnight. The 100% ethanol was replaced sequentially and separately through an isoamyl acetate series (ethanol: isoamyl acetate (3:1), ethanol: isoamyl acetate (1:1), ethanol: isoamyl acetate (1:3), and 100% isoamyl acetate) for 15 min. The samples were then dried using a Critical Point Dryer (HCP-2; Hitachi, Tokyo, Japan) with liquid CO_2_ and coated with platinum using an Ion Sputter (E-1010; Hitachi, Tokyo, Japan). The coated samples were observed and photographed with a scanning electron microscope (S-4800 FESEM; Hitachi, Tokyo, Japan) at 10 keV. All statistical analyses were performed using SPSS v16.0. Independent-Samples T-test was used.

### 4.6. Cryo-Scanning Electron Microscope (Cryo-SEM) Observations

The fresh leaves were taken from wild-type and *35S::PtoCYCD3;3* plants to observe the internal structure of leaves. The fresh samples were cut into approximately 0.2–0.3 cm, then immediately frozen in liquid nitrogen and observed using cryo-scanning electron microscope (cryo-SEM) (Hitachi S-3400N, Tokyo, Japan; Quorum PP3000T, East Grinstead, UK). Data were measured using images generated by Image J. All statistical analyses were performed using SPSS v16.0. Independent-Samples T-test was used.

### 4.7. Determination of Photosynthetic Parameters

The living mature leaves of wild-type and overexpressed plants were selected to detect the photosynthetic parameters under different light intensities. The parameters of the light photosynthetic reactions were measured simultaneously at 25 °C using the Dual-PAM 100 measuring system (Heinz Walz, Effeltrich, Germany). After dark adaptation for 30 min, the electron transfer rate of PSI and PSII were measured under different gradient light intensities of 0, 10, 36, 94, 214, 421, 737, 1178, and 1809 μmol × m^−2^ × s^−1^. The net assimilation rate per unit area was measured at 737 μmol × m^−2^ × s^−1^ light intensity. The net assimilation rate per unit area was multiplied by the total leaf area to obtain the total net assimilation rate. All statistical analyses were performed using SPSS v16.0. Independent-Samples T test was used.

### 4.8. Histological Analysis

The second internode, 10th internode, and 20th internode (counting from the tip) were taken from 4-month-old wild-type and *35S::PtoCYCD3;3* plants. The samples were fixed in FAA (70% ethanol: glacial acetic acid: formaldehyde: glycerol = 18:1:1:1) for 24 h at room temperature and then dehydrated through a graded ethanol series (70%, 80%, 90%, and 100% ethanol; 20 min each step). Ethanol was replaced with ethanol: acetone (1:1) for 20 min and then with pure acetone for 20 min two times. Afterward, the samples were sequentially infiltrated with acetone: resin (2:1) for 3 h and then with acetone: resin (1:1) and acetone: resin (1:3) for 3 h. The samples were replaced with fresh resin for 12 h. After the resin replacement was repeated two times, stem segments were embedded into the resin for 24 h at 60 °C. One µm sections were obtained with a microtome (Leica ultracut R, Germany). The sections were floated on the water, heat-fixed to glass slides, stained with 0.05% toluidine blue, and observed under a Leica DM4000B light microscope. Data were measured on Image J. We measured five times in different quadrant sectors of the cross-sections about the width of cambium-derived tissues. All statistical analyses were performed using SPSS v16.0. Independent-Samples T-test was used.

### 4.9. Yeast Two-Hybrid Assays

The coding sequences of *PtoCDKs* were cloned into the *pGADT7* vector. The coding sequence of *PtoCYCD3;3* was cloned into the *pGBKT7* vector. *Yeast strain* AH109 was co-transfected with *pGBKT7* and *pGADT7* constructs carrying the corresponding genes in accordance with the yeast transformation kit (SK2400-200, Coolaber, Beijing, China) and grown on SD/-Leu/-Trp or SD/-Leu/-Trp/-His/+x-α-gal. No-interacting proteins interchanged vectors. Growth assays were based on Yeast Protocols Handbook (Clontech). Primer sequences were listed in Appendix A.

### 4.10. Homology Modeling

The template crystal structures for PtoCDKE;2, PtoCDKG;3, and PtoCYCD3;3 were identified through BLAST and downloaded from RCSB Protein Data Bank (PDB ID: 5IDN, 6INL, and 4BCM). Homology modeling was conducted in MOE. The protonation state of the protein and the orientation of the hydrogens were optimized by QuickPrep at pH 7 and a temperature of 300 K. First, the target sequence was aligned to the template sequence, and 10 independent intermediate models were built. These different homology models were the result of the permutational selection of different loop candidates and side-chain rotamers. Then, the intermediate model that scored best according to the GB/VI scoring function was chosen as the final model, subject to further energy minimization using the AMBER12/EHT force field.

### 4.11. Molecular Docking

The protein–protein docking protocol in MOE was applied for the docking of Lunasin with proteins. The smaller protein (a smaller number of residues) usually is set as ligand and the bigger one as the receptor. Here, PtoCYCD3;3 was defined as ligand and the others were the receptor, and a multi-stage method was used for generating poses and then ranking them. Starting from a coarse-grained (CG) model to reduce the computational search space, exhaustive sampling was carried out to generate a set of initial poses. The Hopf fibration was used to generate a set of uniformly-distributed rotations, and Fast Fourier Transform (FFT) was used to sample all translations for a given rotation. This was followed by a minimization process that was built around a staged convergence protocol. The conformation of the best ranked was selected as the final (probable) binding mode. Molecular graphics were generated via PyMOL.

### 4.12. Hormone Content Assay

Leaf (from second leaf to fifth leaf) and stem (from first internode to 10^th^ internode) were taken from 4-month-old wild-type and *35S::PtoCYCD3;3* plants. The following method was used to determine the conventional hormone content (IAA, ABA, GA3, TZR): (1) accurately weigh approximately 0.5 g of fresh plant samples, and grind them in liquid nitrogen until crushed; (2) add 5 mL of isopropanol/hydrochloric acid extraction buffer to the powder and shake at 4 °C for 30 min; (3) add 10 mL of dichloromethane and shake for 30 min at 4 °C; (4) centrifuge at 4 °C and 13,000 rpm for 5 min and remove the lower organic phase; (5) avoid light, dry the organic phase with nitrogen, and dissolve it with 200 μL of methanol (0.1% formic acid); and (6) pass the 0.22μm-filter membrane and use HPLC-MS/MS for detection.

The following method was used to determine the BR hormone content: (1) weigh approximately 0.6 g of plant samples accurately and grind it to powder in liquid nitrogen; (2) add 10 mL of 80% methanol pre-cooled at 4 °C and extract at 4 °C for 2 h; (3) centrifuge at 10,000 rpm under 4 °C for 5 min, take the supernatant, pass the Bond Elut pre-packed column, and elute with 3 mL of methanol; (4) add 3 mL of 40 mM ammonium acetate (pH 6.5) to dilute the eluent and pass through a Centricon ultrafiltration tube; (5) take the filtrate through the ultrafiltration tube, pass the Strata-X cartridge, and elute with 3 mL of methanol; (6) dry the methanol with nitrogen and add 200 µl of methanol to dissolve; (7) pass 0.22 μm-filter membrane and conduct HPLC-MS/MS detection. All statistical analyses were performed using SPSS v16.0. Independent-Samples T-test was used.

### 4.13. Accession Numbers

The sequence information of genes from *P. tomentosa* described in this article were deposited into GenBank. Below are the accession numbers of the *P. tomentosa* genes, *PtoCDKA;1* (MT990424), *PtoCDKB1;1* (MT990425), *PtoCDKB1;2* (MT990426), *PtoCDKB2;1* (MT990427), *PtoCDKB2;2* (MT990428), *PtoCDKC;1* (MT990429), *PtoCDKC;2* (MT990430), *PtoCDKD;1* (MT990431), *PtoCDKD;2* (MT990432), *PtoCDKD;3* (MT990433), *PtoCDKE;1* (MT990434), *PtoCDKE;2* (MT990435), *PtoCDKF;1* (MT990436), *PtoCDKG;1* (MT990437), *PtoCDKG;3* (MT990438), *PtoCDKG;4* (MT990439), *PtoCDKG;5* (MT990440), *PtoCYCD3;1* (MT990441), *PtoCYCD3;2* (MT990442), *PtoCYCD3;3* (MT990443), *PtoCYCD3;5* (MT990444), and *PtoCYCD3;6* (MT990445).

## Figures and Tables

**Figure 1 ijms-22-01288-f001:**
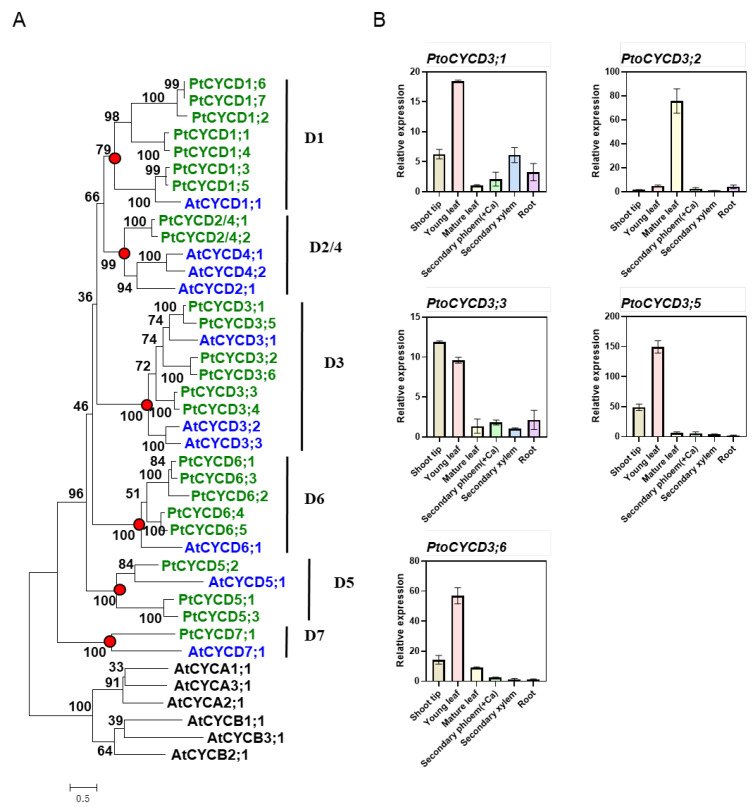
CYCD3 gene identification and expression pattern in *Populus*. (**A**) Phylogenetic tree of *Populus* and *Arabidopsis* D-type cyclin (CYCD) members. (**B**) RT-qPCR analysis of *PtoCYCD3* expression profile in various *Populus* tissues. Ca: Cambium. The expression in each sample was normalized using *P**opulus Actin* (GenBank number XM_002316253) as an internal control. The values are means ±SE, n = 3. Phylogenetic trees were constructed using maximum-likelihood (ML) method in PHYML software with the Whelan and Goldman (WAG) amino acid substitution model. The proportion of invariable sites (I) and gamma distribution (G) parameter was estimated, and the bootstrap replicates were set to 1000.

**Figure 2 ijms-22-01288-f002:**
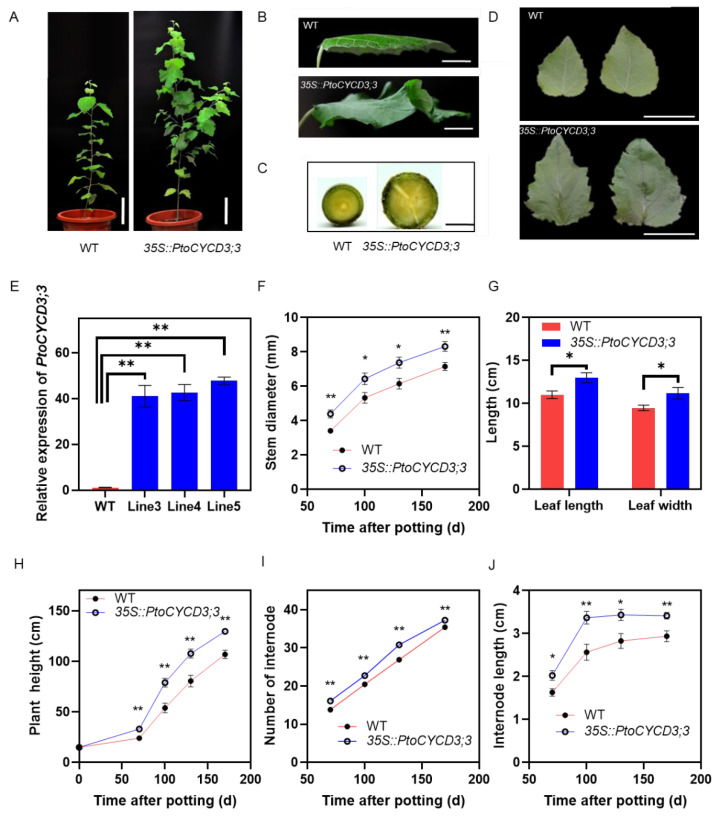
Morphological phenotypes in the overexpression of *PtoCYCD3;3* in *Populus*. (**A**) Whole plants. (**B**,**D**) Leaf of wild-type (WT) and *35S::PtoCYCD3;3* plants. (**C**) Stems. (**E**) Expression levels of *PtoCYCD3;3* in wild-type (WT) plants and three independent lines of *35S::PtoCYCD3;3* plants. The expression in each sample was normalized using *P**opulus Actin* (GenBank number XM_002316253) as an internal control. The bars represent the mean SE of three biological replicates. (**F**) Stem diameter. (**G**) Leaf size. (**H**) Plant height. (**I**) Number of internodes. (**J**) Internode length. Significance as determined by independent sample T-test: * *p* < 0.05, ** *p* < 0.01. The values are means ±SE, n = 10. Ten transgenic lines were used to measure growth parameters. Each transgenic line used one individual. (**A**–**C**) The plant age was at 110 days. (**D**,**E**) The plant age was at 70 days. (**G**) The plant age was at 130 days. Bar, 15 cm (**A**); 2 cm (**B**); 0.5 cm (**C**); 3 cm (**D**).

**Figure 3 ijms-22-01288-f003:**
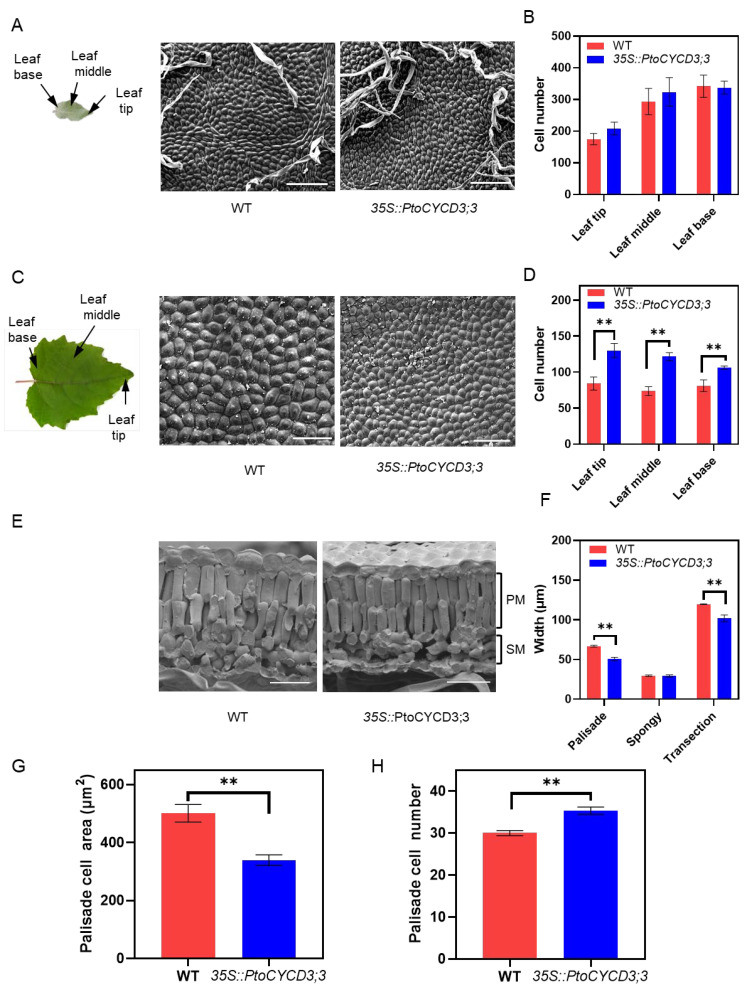
Overexpression of *PtoCYCD3;3* affect leaf development. (**A**) Schematic of a young leaf. Scanning electron micrographs of the adaxial epidermis of wild-type (WT) and *35S::PtoCYCD3;3* young leaves. (**B**) Number of cells per unit leaf area at the tip, middle, and base of WT and *35S::PtoCYCD3;3* young leaves. (**C**) Schematic of a mature leaf. Scanning electron micrographs of the adaxial epidermis of WT and *35S::PtoCYCD3;3* mature leaves. (**D**) Number of cells per unit leaf area at the tip, middle, and base of WT and *35S::PtoCYCD3;3* mature leaves. (**E**) Cryo-scanning electron micrographs of the cross-sections of WT and *35S::PtoCYCD3;3* leaves. PM: palisade mesophyll; SM: spongy mesophyll. (**F**) Palisade mesophyll, spongy mesophyll, and transection width of WT and *35S::PtoCYCD3;3* leaves. (**G**) PM cell area of WT and *35S::PtoCYCD3;3* leaves. (**H**) PM cell number per unit leaf area of WT and *35S::PtoCYCD3;3* leaves. Significance as determined by independent sample T-test: * *p* < 0.05, ** *p* < 0.01. The values are means ±SE, n = 3. Three transgenic lines were used. Bar, 100 µm (**A**,**C**); 40 µm (**E**).

**Figure 4 ijms-22-01288-f004:**
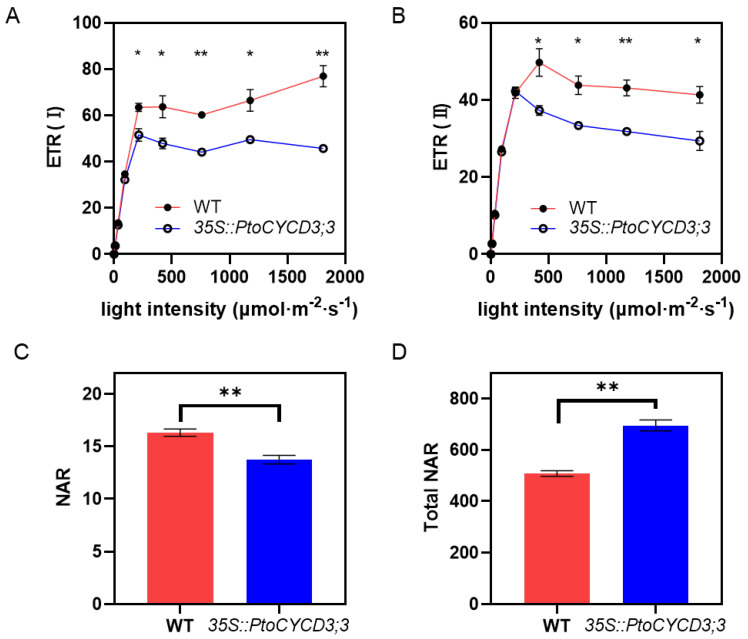
Photosynthesis of wild-type and *35S::PtoCYCD3;3* plants. (**A**,**B**) Electron transfer rate of photosystems I and II under different light intensities. (**C**) Net assimilation rate per unit leaf area. (**D**) Net assimilation rate of total leaf area. Significance as determined by independent sample T-test: * *p* < 0.05, ** *p* < 0.01. The values are means ±SE, n = 3. Three transgenic lines were used.

**Figure 5 ijms-22-01288-f005:**
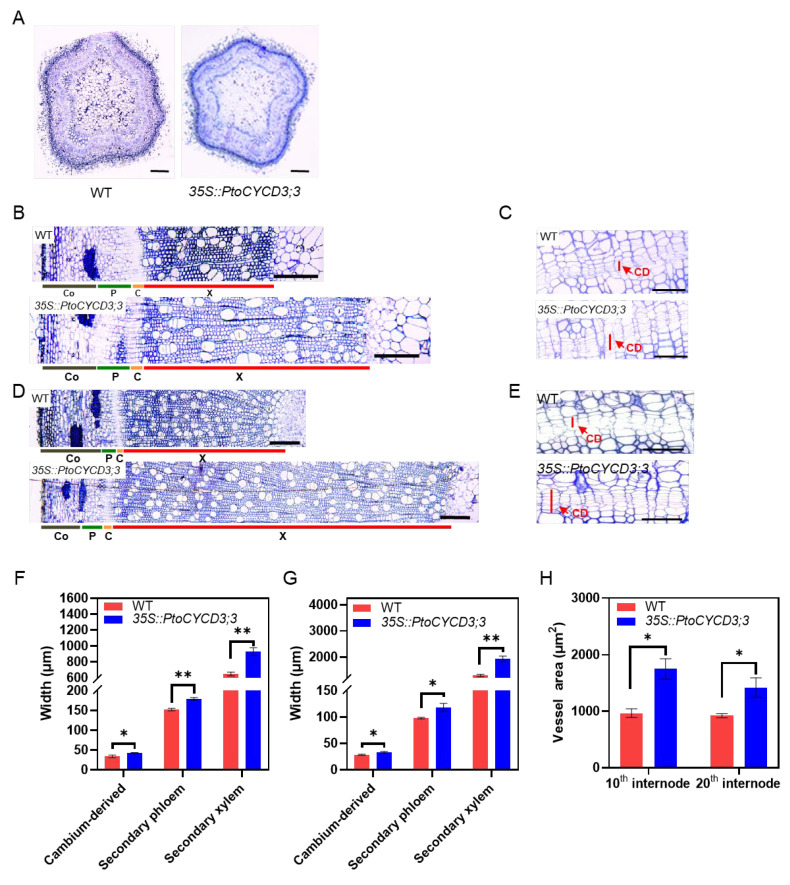
Effects of overexpressing *PtoCYCD3;3* on *Populus* stem development. (**A**,**B**,**D**) Cross-sections of the second, 10th, 20th internode in stems of wild-type (WT) and *35S::PtoCYCD3;3* plants. (**C**,**E**) Cambial-derived tissue of the 10th and 20th internode in stems of WT and *35S::PtoCYCD3;3* plants. (**F**) Width of cambial-derived tissue, secondary phloem, and secondary xylem of the 10th internode in stems of WT and *35S::PtoCYCD3;3* plants. (**G**) Width of cambial-derived tissue, secondary phloem, and secondary xylem of the 20th internode in stems of WT and *35S::PtoCYCD3;3* plants. (**H**) Vessel area of the 10th and 20th internode in stems of WT and *35S::PtoCYCD3;3* plants. Co: cortex, P: phloem C: cambial, X: xylem, CD: cambial-derived tissue. Significance as determined by independent sample T-test: * *p* < 0.05, ** *p* < 0.01. The values are means ±SE, n = 3. Three transgenic lines were used. Bar, 100 µm (**A**); 200 µm (**B**–**E**).

**Figure 6 ijms-22-01288-f006:**
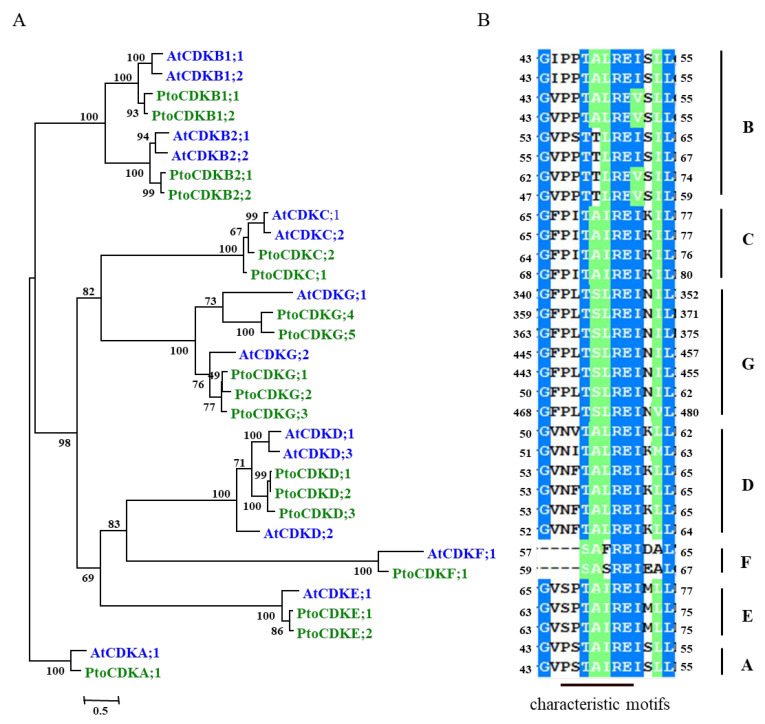
Identification and sequence analysis of members of the cyclin-dependent kinases (CDK) family in *Populus*. (**A**) Phylogenetic tree of *Populus* and *Arabidopsis* CDK members. (**B**) Characteristic motifs of different CDK members.

**Figure 7 ijms-22-01288-f007:**
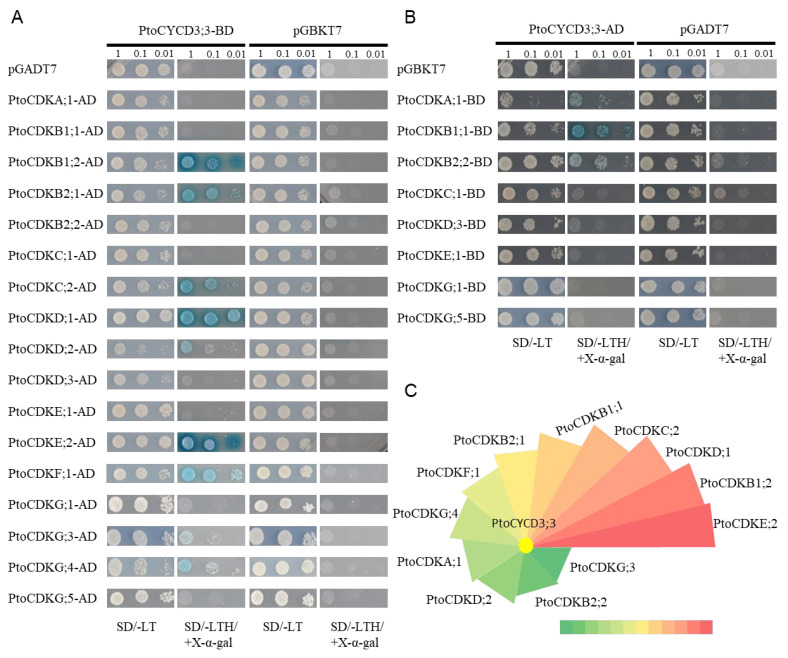
Protein–protein interactions of PtoCYCD3;3 with CDKs. (**A**,**B**) Yeast cells were co-transfected with pGBKT7 and pGADT7 constructs carrying the corresponding genes and grown on SD/-Leu/-Trp or SD/-Leu/-Trp/-His/+x-α-gal. pGADT7 + PtoCYCD3;3-BD, PtoCDKs-AD +pGBKT7, pGBKT7 + PtoCYCD3;3-AD, PtoCDKs-BD + pGADT7, and pGADT7 +pGBKT7 as negative control. (**C**) Interaction strength of PtoCYCD3;3 with CDKs. The color gradient (green-yellow-red) represents the interaction strength, ranging from the minimum to the maximum.

**Figure 8 ijms-22-01288-f008:**
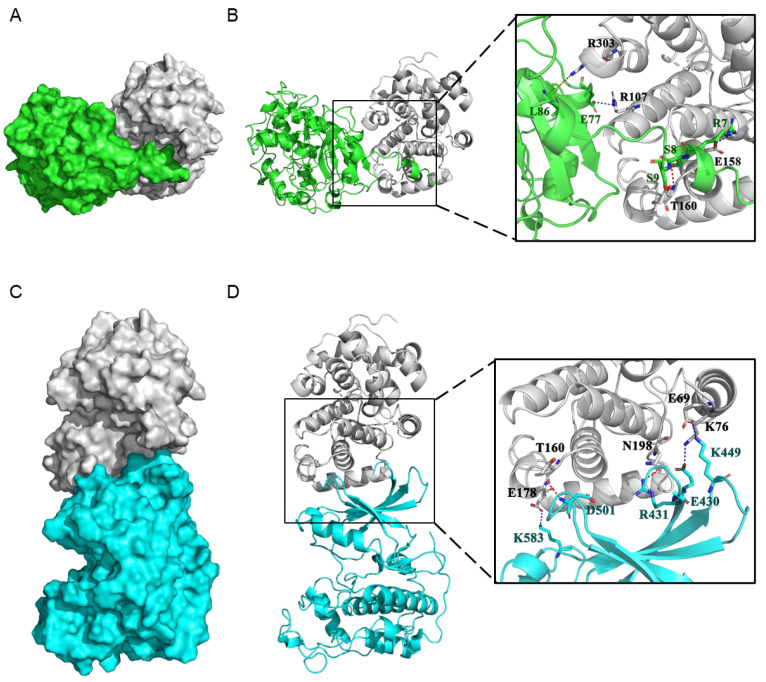
Protein–protein docking of PtoCYCD3;3 with PtoCDKE;2 and PtoCDKG;3. (**A**) Surface binding model between PtoCDKE;2 and PtoCYCD3;3. The surface of PtoCDKE;2 is colored in green. The surface of PtoCYCD3;3 is colored in gray90. (**B**) Detail binding model between PtoCDKE;2 and PtoCYCD3;3. The backbone of PtoCDKE;2 is depicted as green cartoon. The residues in PtoCDKE;2 are colored in green. The backbone of PtoCYCD3;3 is depicted as gray90 cartoon. The residues in PtoCYCD3;3 are colored in gray90. The red dashes represent hydrogen bond interaction, and the blue dash represents salt bridge. (**C**) Surface binding model between PtoCDKG;3 and PtoCYCD3;3. The surface of PtoCDKG;3 is colored in cyan. The surface of PtoCYCD3;3 is colored in gray90. (**D**) Detail binding model between PtoCDKG;3 and PtoCYCD3;3. The backbone of PtoCDKG;3 is depicted as cyan cartoon. The residues in PtoCDKG;3 are colored in cyan. The backbone of PtoCYCD3;3 is depicted as gray90 cartoon. The residues in PtoCYCD3;3 are colored in gray90. The red dashes represent hydrogen bond interaction, and the blue dash represents salt bridge.

## Data Availability

Not applicable.

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
