# Peer review of "Overexpression of PtoCYCD3;3 Promotes Growth and Causes Leaf Wrinkle and Branch Appearance in Populus"

_ijms, 2021, doi:10.3390/ijms22031288_

Round 1
Reviewer 1 Report
In this manuscript, the authors present a study on D-type cyclin genes in Populus tomentosa. The authors present a gene functional analysis by overexpressing PtCYD3 gene under the constitutive 35S promoter to understand this gene role in Populus. The authors show that transgenic overexpression of PtoCYCD3;3 causes increased growth specially in poplar leaves and vascular cambium. The authors also show cell divisions at the adaxial side of the leaves are stimulated, as well as at the cambial zone growth in the stems. Together with protein interaction analysis with Y2H these results constitute the most compelling and complete analysis in the manuscript. However, this is also the main weakness, given that the documented phenotypic analysis shows no multiple transgenic lines and it is not clear to which or how many lines were used in the experiments. The results description feels at point preliminary and would also benefit from some re-writing to create an interesting storyline. Results are interesting and the manuscript could be improved if writing would be improved, which is really worth the effort to intertwine the rational followed from one experiment to the other. Some comments/suggestions follow:
- The authors mention (Line 332) “results showed that PtoCYCD3;3 could promote the differentiation of stem cambium cells and leaf development and increase biomass by enhancing plant vegetative growth.” Suggestion of increased periclinal and anticlinal cell divisions at the cambium is quite interesting given the low CK accumulation in the stems (Fig. S6), which is intriguing, but not undescribed when using constitutive 35S promoter. This was observed recently (e.g. Front. Plant Sci. 11:601858. doi: 10.3389/fpls.2020.601858), and attributed it to perhaps differences in CK effects on anticlinal/periclinal cell divisions, which is quite opposite to earlier description on lower CKs and reduced periclinal cell divisions at the cambial region of Populus (PNAS 105: 20032-20037. doi:10.1073/pnas.0805617106; or higher as in Curr. Biol. 2014; 24: 2053-2058, 10.1016/j.cub.2016.05.053. and how it seems to operate in Arabidopsis seedlings). The idea of a negative feedback loop controlling cell division at the cambium and in branching is very compelling and, there are previously described examples in common between Arabidopsis and Populus secondary growth involving auxin and cytokinin (e.g. Plant J. 2013 75(4):685-98. doi: 10.1111/tpj.12231 and Molecular Plant 2014, 7(6): 1006-1025. Doi: 10.1093/mp/ssu051). The authors could improve their discussion further on their results given their hormones quantification. These are examples of related feedback mechanisms in literature, authors can look for others they find more suitable.
- Still about the comment above, it is uncertain whether the feedback loop operates in this case. Exogenous application of hormones in the transgenics and wild-type could further elucidate on the mechanistics. It is not required to show these experiments here though. It is quite interesting to raise the point in the discussion.
- Regarding results shown in Fig 1. Could the authors please compare your results in Pto to previously described for Populus (Plant Physiol. 2007;145(4):1558-1576. doi:10.1104/pp.107.104901). Please provide details on the bootstrap values and tree construction in the caption. Include gene IDs or protein IDs in the supplementary file relatively to sequences used.
- The authors should make an effort to improve some details in the descriptions of the captions. For example, it is not clear which line was used for growth parameters (Fig. 2). The caption says “The values are means ± SE, n = 10. “but which transgenic line was used? Are these mean values of the 3 transgenic lines together? How many individuals per transgenic line were used? Authors should show each individual line results in the analysis perhaps in supplementary and clarify in the main text and main figures what the values represent in terms of biological replicates per transgenic line. To me it would be important to show how the phenotypes relate to the level of expression in the several transgenic lines to be able to draw conclusions.
- One curious aspect that is visible and highlighted is that there seems to be branching in the overexpressors, how did branching affect secondary growth in the more mature internodes? Were all overexpressor plants branched? Maybe measuring strigolactones (SL) besides auxins would make sense in this case. Incorporating work on SL and branching and secondary growth would enrich the discussion (e.g. Proc Natl Acad Sci U S A. 2011 108(50): 20242–20247; 10.1073/pnas.1111902108). Do you think perhaps increased shoot weight might have affected cambial growth?
- How was the width of cambium and cambial-derivatives measured relatively to the whole section of the stem? Were measurements taken in different quadrant sectors of the cross sections? How many measurements? Provide details either in captions/methods.
- There is a poor transition between your results from PtoCYCD3;3 and the CDKs, could the authors please introduce some transition sentence for the rational of searching for the CDKs after observing that CYCD3,3 affected secondary growth.
- The manuscript bit on the protein-protein docking in silico analysis for me adds little to the findings and could perhaps be transferred to supplementary?
Minor comments
Line 23: This sentence reads strange “These genes have been amplified in Populus tomentosa by using PtCYCD3 genes as a template”. What is meant? If this is the case, perhaps substitute by PtCYD3 genes expression was examined in Populus tomentosa. If full length sequences were isolated from genomic or cDNA or libraries?
Line 59: add in which species to the sentence. Reference your work, also further down the text e.g. CDKs in Arabidopsis, surely have a reference. There are many instances in the text where references are missing. Needs polishing in this respect.
Line 126: Please could the authors correct where it says “70, 100, 130, and 170 days after cutting.” It should be “after potting”, right?
Line 402 Again, please provide further details, on the sequences isolated. When the authors say genes were isolated, was it full length? Were 5’ 3’ UTRs included?
Line 424 “transgenic and wild-type plants were grown in the natural environment for 6 months.” Please provide details, it is not clear what is meant by natural environment growth. How many trees were grown from each transgenic line, the results report one transgenic line or several? The same for the 10 biological replicates where growth parameters were taken from.
- Please clarify statistics in methods, clarify which statistical test was used.
- How was “secondary xylem, and secondary phloem (including cambium, 20th internode counting from the tip)” harvested? Was it scraped? Please clarify in methods. Were same batches of samples divided for several analysis (qPCR, hormone). Please clarify in the text.
- Could the authors show that PtoActin (GenBank number XM_002316253) housekeeping gene is good enough to use in this species. Given that you normalize all your expression results to one single reference gene perhaps better to reference it. Also refer to how many biological replicates and which transgenic line was used.
-Please could the authors reference the methods where applicable.
Reviewer 2 Report
Abstract needs work.
D-type cyclin (cyclin D, CYCD), combined with cyclin-dependent kinases(CDKs), participates in the regulation of cell cycle G1/S transition and plays an important role in cell division and proliferation. CYCD is believed to affect growth and development of herbaceous plants, such as Arabidopsis thaliana, by regulating the cell cycle process. This study was undertaken to examine the potential role of CYCD in woody plants (e.g., poplar). Phylogenetic analysis showed that in Populus trichocarpa, CYCD3 genes expanded to six members, namely PtCYCD3;1–6. PtoCYCD3;3 showed the highest expression in the shoot tip, secondary phloem (including cambium), and the higher expression in young leaves among all members. Therefore, this gene was selected for further study. The overexpression of PtoCYCD3;3 in plants demonstrated obvious morphological changes during the observation period. The leaves became enlarged and wrinkled, the stems thickened and elongated, and multiple branches were formed by the plants. Anatomical study showed that in addition to promoting the differentiation of cambium tissues and the expansion of stem vessel cells, PtoCYCD3;3 facilitated the division of leaf adaxial epidermal cells and palisade tissue cells. Yeast two-hybrid experiment exhibited that 12 PtoCDK proteins could interact with PtoCYCD3;3, of which the strongest interaction strength was PtoCDKE;2, whereas the weakest was PtoCDKG;3. Molecular docking experiments further verified the force strength of PtoCDKE;2 and PtoCDKG;3 with PtoCYCD3;3. In summary, these results indicated that the overexpression of PtoCYCD3;3 significantly promoted the vegetative growth of Populus, and PtoCYCD3;3 may interact with different types of CDK proteins to regulate cell cycle processes.
Keywords: CYCD3
44 Don't start sentence with pronoun.
45 CYCDs in accordance with the low sequence homology with animal CYCD --> It is NOT BECAUSE OF LOW sequence homology // it is BECAUSE of sequence homology.
Misleading: "These genes have been amplified in Populus tomentosa by using PtCYCD3 genes as a template." Template is misused here ... At is not a template for populus nor is populus for At reverse ...
61 overexpressed Nicta;CYCD3;4 // define abbrev or use common name tobacco
68 cycd3;1 substantially decreased [17]. --> in plants of mutant cycd3;1 was substantially decreased [17].
69 While --> Although
71 and few kinds of studies have been reported --> and few studies have reported
72 In woody plant stems, the activity of cambium cell is --> In woody plant stems, the activation of cambium cells is
74 Therefore, it has a great theoretical and application value for researching the function of CYCD3 in woody plants. /// "it" = what??? bad pronoun
77 Populus tomentosa is one of the widely spread cultivated economic tree species in China --> Populus tomentosa is a widespread cultivated tree species of economic importance in China
76 A relatively complete system of genetic transformation has been developed in P. tomentosa, which makes this species one of the most significant tree species for forest genetic research.
78 Six CYCD3 genes were homologously 78 searched in Populus trichocarpa in this study. --> Six homologous CYCD3 genes were examined in this study to to document their expression and role in Populus trichocarpa
80 P. tomentosa by using populus genes as a template. --> using P. tomentosa as a template.
83 the cambium cell --> cambium cell
84 In this study, the effects of PtoCYCD3;3 on Populus growth and development was examined and CDK family members that interact with PtoCYCD3;3 85 were analyzed, laying a foundation for further exploration of the molecular mechanism of cyclins in woody plants.
90 via homologous search in the P. --> via searching homologies in the P.
93 Arabidopsis mainly clustered --> Arabidopsis clustered [omit mainly]
95 matched --> closely resembled the pattern of
101 while ---> whereas [while is a comparison of time]
104 Six CYCD3 genes were homologously searched in P. trichocarpa. --> Six CYCD3 genes were examined for homologous patterns of nucleotides in P. trichocarpa.
105 Five PtoCYCD3 genes were identified and amplified in P. tomentosa by using them as a template. Among them ---> who is 'them'??
126 The plant height of the overexpressed plants was significantly higher than that of the wild-type plants during the growth process from 70 days to 170 days (Figure 2H).
NOTE: It seems that the amount of over-expression should be stated. What is cut-off? If p<0.05 or other number, please state. Any time that authors used "significantly higher", the p value should be stated or you should have a statement of what is meant numerically by significant.
133 "over expression plants" is not clear as to what is meant
151 "The plant height of the overexpressed plants was significantly higher than that of the wild-type plants during the growth process from 70 days to 170 days (Figure 2H)....." --> add "That" to sentence "That the plant height of the overexpressed plants was significantly higher than that of the wild-type plants during the growth process from 70 days to 170 days (Figure 2H) may be due to the increase in leaf cell number or the increase in cell area."
154 "no substantial difference" has no inherent meaning. If this statistically significant and if so, state p level is meant by " statistically significant" especially if you change definitions. Otherwise just state, that values of p<0.05 are significant. Define if you have other criteria.
189 "In green plants, palisade tissue is the main place for leaf photosynthesis." citation????
213 wild-type plants was observed --> wild-type plants were observed
229 encouraged by --> increased by
Figure 5: Improvement to the color is needed in this figure. Most of the micrographs in this figure have a background that that is pink rather than a neutral white tint. Using Photoshop or ImageJ or other graphic program should be able to normalize the tint to a neutral white. This is a color temperature effect. Figure E does not display this effect but A, B, C, and D do. These should routinely be normalized to white so that the colors are accurately portrayed in the journal.
277 while --> whereas
319 an decreased --> decreased
322 These finding --> These findings
322 similar manner -- manner similar
224 the decrease in the proliferation of chloroplasts --> a decrease in the number of chloroplasts
330 A significant increase in stem thickness and the expansion of vessel cells in overexpressed PtoCYCD3;3 poplars was apparently related to vigorous cambium cell activity that may have enhanced the differentiation and development of secondary xylem and secondary phloem.
337 CDK --> CDKs
458 S-3400N, Tokyo, Japan; Quorum PP3000T, East Grinstead, UK. Data were measured using images generated by ImageJ.
462 the condition of different light intensities --> different light intensities.
Reviewer 3 Report
The manuscript entitled: “Overexpression of PtoCYCD3;3 promotes growth and causes leaf wrinkle and branch appearance in Populus” by Guan et al. describes the results of the PtoCYCD3;3 overexpression pointing to important functions of the D-type cyclin encoded by this gene. Moreover, the protein interactors of the CYCD3;3 were identified confirming the significant role of the cyclin in cell cycle regulation. In general, results of the performed experiments convincingly explain the mechanisms underlying the morphological changes caused by overexpression. However, the design and interpretation of the qPCR assays should be revised as well as some minor issues related to experiments description and clarity of data presentation should be addressed.
Major comments:
- The qPCR experiments are applied to compare expression levels of five different genes encoding PtoCYCD3;1, 2, 3, 5, 6. Such comparisons are performed in six tissues/organs. The qPCR technique is not supposed to be applied for this type of analyses since results (relative expression levels) depend not only on the copy number of template cDNA but also on amplification efficiency which varies a lot depending on the amplified fragment and is determined mainly by primers performance (and primers are different for each amplicon). In the case of the qPCR experiments presented in the manuscript, each of the five amplicons is most probably amplified with different efficiency and therefore the results are not reliable. Instead, the relative expression levels should be established and compared between tissues/organs for each gene separately. To compare the expression level between the genes another technique could be applied, for example Northern-Blot (if the probe common for each gene could be designed) or (the best option) – the digital PCR.
- It is not clear whether the data shown on six different charts can be compared to conclude in which tissue/organ the CYCD3;3 (or any other gene) has the highest expression. This would be possible only if all date were obtained in the same experiment (in such a case the units of the Relative expression level would be the same).
- The quantification method applied by authors is based on the assumption that each amplicon has maximal efficiency of 2 (or 100%). Much precise quantitative analysis would be performed if amplification efficiency was established for each gene.
Minor comments:
- Abstract (lines 23-24) – “genes have been amplified in Populus tomentosa by using PtCYCD3 genes as a template” – the word “template” is very misleading. Since authors write about amplification it may suggest that P. trichocarpa DNA was used.
- In the overexpression results presented in Fig.2 the plant age is not clear in some panels (A, B, C, D, E, G).
- Lines 126 and 134 – the word “cutting” should be probably replaced by “potting”.
- It is not clear how the leaf area was measured and which values were obtained for WT and overexpressing line (only the difference is given in line 138).
- In lines 168 – 171 the use of “width” and “length” is confusing and should reconsidered.
